# Immune Checkpoint Blockade Combined with AbnobaViscum^®^ Therapy Is Linked to Improved Survival in Advanced or Metastatic Non-Small-Cell Lung Cancer Patients: A Registry Study in Accordance with the ESMO Guidance for Reporting Real-World Evidence

**DOI:** 10.3390/ph17121713

**Published:** 2024-12-18

**Authors:** Friedemann Schad, Anja Thronicke, Ralf-Dieter Hofheinz, Reinhild Klein, Patricia Grabowski, Shiao Li Oei, Hannah Wüstefeld, Christian Grah

**Affiliations:** 1Network Oncology Registry, Research Institute Havelhöhe, Kladower Damm 221, 14089 Berlin, Germany; shiaoli.oei@havelhoehe.de; 2Interdisciplinary Oncological Centre, Hospital Havelhöhe, Kladower Damm 221, 14089 Berlin, Germany; patricia.grabowski@havelhoehe.de; 3Mannheim Cancer Center, Mannheim University Hospital, Theodor-Kutzer Ufer 1-3, 68167 Mannheim, Germany; ralf-dieter.hofheinz@medma.uni-heidelberg.de; 4Department of Internal Medicine II, University Hospital Tübingen, Otfried-Müller Straße 10, 72076 Tübingen, Germany; reinhild.klein@med.uni-tuebingen.de; 5Charité Fatigue Centrum, Charité—Universitätsmedizin Berlin, Augustenburger Platz 1, 13353 Berlin, Germany; 6Lung Cancer Center, Hospital Havelhöhe, Kladower Damm 221, 14089 Berlin, Germany; hannah.wuestefeld@havelhoehe.de (H.W.); christian.grah@havelhoehe.de (C.G.)

**Keywords:** PD-1 inhibitor, PD-L1 inhibitor, survival, abnobaViscum^®^ therapy, non-small-cell lung cancer, lung cancer

## Abstract

**Background:** Recent advancements in cancer treatment have shown the potential of immune checkpoint blockade (ICB) plus *Viscum album* L. therapy in improving survival rates for patients with advanced or metastatic non-small-cell lung cancer (NSCLC). The objective of this study was to investigate factors associated with improved survival in NSCLC patients treated with a combination of ICB and abnobaViscum^®^. **Methods**: Patients with advanced or metastatic NSCLC from the accredited Network Oncology registry were included in this real-world data study adhering to ESMO-GROW criteria with ethics approval. Survival outcomes were compared between patients receiving ICB therapy alone versus those receiving combinational ICB plus abnobaViscum^®^ therapy using Kaplan–Meier and multivariable Cox proportional hazard analysis. **Results**: Among 300 patients (median age 68 years; male/female ratio 1.19), 222 received ICB alone (CTRL group) and 78 received combinational therapy (COMB group). Overall survival was significantly prolonged in the COMB group by 7 months compared to CTRL (13.8 months vs. 6.8 months, *p* = 0.005) with a survival rate of 16.5% in the COMB group vs. 8.0% in the CTRL group. In programmed death-ligand 1 positive (≥1%) patients treated with first-line ICB, the addition of abnobaViscum^®^ reduced the adjusted hazard of death by 75% (aHR: 0.25; 95%CI: 0.11–0.60, *p* = 0.02). **Conclusions**: The addition of abnobaViscum^®^ to ICB is significantly associated with improved survival in patients with advanced or metastatic NSCLC patients, irrespective of age, stage, Eastern cooperative oncology group status, surgery, or radiation. Potential mechanisms include immune modulation, reduced primary ICB resistance, and tumor microenvironment modifications. The findings warrant further validation in randomized controlled trials or registry-based randomized controlled trials. Trial registration: The study was registered (DRKS00013335).

## 1. Introduction

Lung cancer is the leading cause of cancer-related deaths globally, accounting for one in five (18.7%) cancer deaths [1]. Non-small-cell lung cancer (NSCLC) comprises approximately 85% of all lung cancer cases while small-cell lung cancer (SCLC) represents the remaining 15% [1]. Despite advancements in diagnosis and therapeutic interventions, the five-year survival rates for metastatic NSCLC remain below 20% [2]. Key risk factors include tobacco use, responsible for 90% of cases, as well as radon, asbestos, air pollution, genetic predispositions, and medical conditions such as chronic lung diseases or dietary deficiencies [3,4].

Advancements in immune checkpoint blockade (ICB) targeting programmed cell death protein 1 (PD-1), programmed death-ligand 1 (PD-L1), or cytotoxic T-lymphocyte antigen 4 (CTL-A4) have revolutionized NSCLC treatment [5,6]. Normally, interactions between receptors like PD-1 on T-cells and their ligand, such as PD-L1 on tumor cells, create an “immune checkpoint” that suppresses T-cell activity, preventing immune-mediated destruction of cancer cells [6]. Tumors leverage this pathway to evade immune surveillance. ICB restores T-cell activity, reactivating T-cells and enhancing anti-tumor immunity [7]. The combinations, timing and duration of current PD-1 inhibitors (e.g., pembrolizumab, nivolumab), PD-L1 inhibitors (e.g., durvalumab, atezolizumab), and CTL-4A inhibitors (e.g., ipilimumab) are continually evolving due to rapid clinical advancements [8,9,10,11,12,13,14,15]. Emerging strategies include first-line and neo-adjuvant applications of ICB and the development of novel checkpoint inhibitors targeting Lymphocyte Activation Gene-3, T-cell Immunoglobulin and Mucin Domain-Containing Protein 3, T-cell Immunoreceptor with Ig and ITIM Domains, and B- and T-Lymphocyte Attenuator [6,10,11,15]. Despite these advancements, challenges like resistance and limited efficacy in certain cancer subgroups remain to be addressed [7].

*Viscum album* L. extracts (VA, European white-berry mistletoe) have shown potential as a complementary therapy for NSCLC [16,17]. Studies suggest that VA, when added to ICB, was associated with improved overall survival in advanced or metastatic NSCLC patients without increasing adverse events [17,18,19,20]. VA’s bioactive compounds, including viscotoxins and mistletoe lectins (MLs), may be particularly relevant for lung cancer therapy due to their cytotoxic and immune-modulating effects [21,22,23,24,25]. Cysteine-rich viscotoxins disrupt cell membranes, inducing permeability or lysis, while MLs inhibit protein synthesis in ribosomes and promote apoptosis [26]. Additionally, MLs enhance immune responses by stimulating macrophages, T-cells, and natural killer cells, and inducing cytokine release (e.g., TNF-a, IL-1, or IL-6) [27]. Other components of VA, like flavonoids and phenolic compounds, modulate oxidative stress and inhibit angiogenesis [28,29]. These synergistic mechanisms make VA a promising complementary therapy improving the efficacy of ICB therapies in NSCLC management.

The aim of this study was to assess the overall survival of patients with advanced or metastatic NSCLC undergoing standard oncological immunotherapy, both with and without abnobaViscum^®^ therapy. To achieve this objective, we utilized real-world data (RWD) from a registry-based source, which was particularly suitable for its relevance and ability to provide a comprehensive picture of real clinical practice. In recent years, the use of RWD has gained significance, particularly in generating evidence for clinical studies [30]. This methodology facilitates the identification of relevant subgroups, such as elderly NSCLC patients with a poor performance status or those with multiple comorbidities, from real-world clinical settings [31]. This approach generates targeted evidence essential for evaluating the effectiveness of ICB, both with and without abnobaViscum^®^ therapy.

## 2. Results

### 2.1. Baseline Characteristics

A total of 300 patients with advanced or metastatic NSCLC who received PD-1/PD-L1 inhibitors as part of their standard of care, as documented in the Network Oncology registry, were included in the study. Of these, 222 patients (74%) were treated with immune checkpoint inhibitors alone, without the addition of abnobaViscum^®^ therapy (control group, CTRL), while 78 patients (26%) received PD-1/PD-L1 inhibitors in combination with abnobaViscum^®^ therapy (combinational group, COMB) (see flowchart, Figure 1). In total, 300 patients were included in the Kaplan–Meier survival analysis and 121 patients were enrolled in the adjusted multivariate Cox regression analysis. The latter group consisted of patients with PD-L1 positive (≥1%) NSCLC who were treated with first-line PD-1/PD-L1 inhibitors only; see Figure 1.

No significant differences were observed between the two groups regarding gender, histology, tumor stage, and surgery. The median age of the total cohort was 68 years (interquartile range of 62–76 years). Participants from the COMB group were on average 0.8 years younger than those in the CTRL group, though this difference was not statistically significant; see Table 1. The sex ratio (male/female) was 1.19. The most common histological subtype of NSCLC was non-squamous cell carcinoma, accounting for 63% (*n* = 189) of cases, followed by squamous cell carcinoma at 30% (*n* = 90), as shown in Table 1. In 7% of patients (*n* = 21), the NCLC diagnosis was not further specified due to the nature of RWD. In the COMB group, the percentage of patients with non-squamous cell carcinoma was slightly lower (61.5%) compared to the CRTL group (63.5%), but the differences between the groups were not statistically significant.

### 2.2. Tumor Markers

No significant differences in the molecular markers were noted between the two groups, except for PD-L1 status, as shown in Table 2. In the CTRL-group, there was a 19.3% higher proportion of patients with a positive PD-L1 status and a 20.1% higher proportion with a PD-L1 ≥50% tumor proportion score (TPS) compared to the COMB group, and these differences were statistically significant. PD-L1 status was available for 82.3% (*n* = 247) of the patients assessed. Meanwhile, the documentation of known B-rapidly accelerated fibrosarcoma (BRAF) mutations, epidermal growth factor receptor (EGFR) exon 18–21 mutations, and anaplastic lymphoma kinase (ALK) translocations varied from 49.3% to 57%, as detailed in Table 2. Regarding stage IV NSCLC, molecular status was recorded for 184 (61.3%) of the 238 patients.

### 2.3. Oncological Treatment

Almost 60% of all enrolled patients received first-line PD-1/PD-L1 therapy. PD-1 inhibitors were the most commonly used (92%) compared to PD-L1 inhibitors (7%); see Table 3. There was no significant difference in the use of PD-1 or PD-L1 inhibitors between the two groups. The CTRL group had 13.6% more patients who received first-line treatment, a difference that approached statistical significance. In total, 11% of enrolled patients received lung radiation and 9% received brain radiation. Additionally, 11% of patients underwent surgery and 4% received chemotherapy. While 9.3% more patients in the COMB group underwent surgery compared to the CTRL (*p* = 0.03), no significant differences were observed between the two groups regarding radiation or chemotherapy.

### 2.4. Characterization of Combinational PD-1/PD-L1 Inhibitor and abnobaViscum^®^ Therapy

The median duration of PD-1/PD-L1 therapy was 135 days (IQR 48–242 days) or approximately 4.42 months (IQR 1.6–7.9 months). In contrast, abnobaViscum^®^ therapy lasted a median of 242 days (IQR 48–464 days) or approximately 7.9 months (IQR 1.6–15.2 months).

Among the patients who received PD-1 inhibitors (either pembrolizuamb or nivolumab), 50.7% received concomitant intravenous abnobaViscum^®^ therapy, 36% concomitant subcutaneous therapy, and 13.3% concomitant intratumoral abnobaViscum^®^ therapy; see Figure 2.

Among the patients who received PD-L1 inhibitors (either atezolizumab or durvalumab), 25% received concomitant intravenous and 75% received concomitant subcutaneous abnobaViscum^®^ therapy; see Figure 2. In most cases, intravenous, subcutaneous, or intratumoral abnobaViscum^®^ therapy was abnobaViscum^®^ fraxini (fraxini = ash tree) extract, which was applied to the patients at varying doses and in different forms; see Table 4. Of all the patients, 37.2% patients received intravenous abnobaViscum^®^ fraxini with VA from other producers; see Table 4. Combinations with VA from other producers were seen in 19.2% of cases for subcutaneous abnobaViscum^®^ fraxini and 7.7% in intratumoral applications; see Table 4. Other abnobaViscum^®^ applications included subcutaneous abnobaViscum^®^ abietis (fir tree), abnobaViscum^®^amygdali (almond tree), or abnobaViscum^®^quercus (oak tree) extracts, which were combined with VA from other producers in 4.2% of all cases.

### 2.5. Overall Survival of Advanced or Metastatic NSCLC Patients Treated with PD-1/PD-L1 Inhibitors Plus Add-On Abnobaviscum^®^

A Kaplan–Meier survival analysis was performed for three hundred patients. Regarding the overall survival, the COMB treatment (PD-1/PD-L1 inhibitors + abnobaViscum^®^ therapy) demonstrated a survival advantage compared to the CTRL group (PD-1/PD-L1 inhibitors without VA), as illustrated in Figure 3. The median survival in the COMB group was 13.8 months (95%CI: 9.2–22 months), which is seven months longer than the median survival in the CTRL group which was 6.8 months (95%CI: 4.9–10.4 months). The log-rank test showed a significant difference (X^2^ = 7.9, *p* = 0.005), as detailed in Table 5. The three-year survival rate was 16.5% in the COMB group, which is twice as high as the 8% three-year survival rate in the CTRL group.

### 2.6. Add-On Abnobaviscum^®^ Therapy Is Associated with Reduced Hazard of Death in PD-L1 Positive (≥1%) NSCLC Patients Treated with First-Line PD-1 Inhibitors

The adjusted multivariate Cox proportional hazard analysis on PD-L1 positive (≥1%) patients with advanced or metastatic NSCLC receiving first-line anti-PD-1 treatment indicated a statistically significant 75% reduction in the adjusted hazard of death (adjusted hazard ratio—aHR: 0.25, 95%CI: 0.11–0.60, *p* = 0.002; adjusted *p* = 0.015) when abnobaViscum^®^ therapy was added, as outlined in Table 6. This effect was independent of gender, age, Eastern Cooperative Oncology Group (ECOG) performance status, tumor stage, surgery, brain radiation, or PD-L1 TPS.

## 3. Discussion

In this RWD study, we assessed the effectiveness of PD-1/PD-L1 inhibitor therapy when combined with abnobaViscum^®^ therapy in cancer patients. Our results indicate that patients with advanced or metastatic NSCLC who received PD-1/PD-L1 inhibitors in combination with abnobaViscum^®^ therapy experienced improved survival compared to patients receiving PD-1/PD-L1 inhibitors alone. Furthermore, patients with a PD-L1 positive (≥1%) NSCLC tumor treated with first-line PD-1 inhibitors in combination with abnobaViscum^®^ therapy had better survival outcomes compared to those treated with PD-1 inhibitors without the add-on abnobaViscum^®^ therapy. This effect was maintained regardless of gender, age, tumor stage, ECOG performance score, oncological treatment, or PD-L1 TPS.

The higher percentage of patients in the control group with positive PD-L1 (≥50% TPS) status compared to the combination group indicates that the combination ICB + abnobaviscum^®^ therapy group had worse starting conditions for survival analyses. Thus, the observed survival advantage in the COMB group is likely attributable to the addition of abnobaviscum^®^ therapy since PD-L1 status was adjusted for in multivariable Cox regression analysis. Additionally, the smaller proportion of patients in the control group receiving surgery was also accounted for in our multivariate Cox regression analysis. Thus, the survival advantage with additional abnobaviscum^®^ therapy in ICB-treated patients was confirmed irrespective of PD-L1 status, surgery, or other confounders.

Clinical evidence is increasingly demonstrating the survival benefits of VA in cancer patients [32]. Numerous systematic reviews, meta-analyses, and both clinical and RWD studies suggest a positive impact of VA extracts on survival outcomes [16,33,34,35,36,37,38,39]. The underlying cause of the additive effect of VA in the present study on the survival in patients with NSCLC treated with PD-1/PD-L1 inhibitors remains open. However, evidence indicates that VA does not interfere with the expression of PD-ligands in lung, breast, colon, and prostate cancer cell lines [40]. On the other hand, in breast cancer spheroids, VA extracts from the spruce tree significantly reduced PD-L1 expression [41]. Moreover, VA extracts have been shown not to hinder the effect of PD-1 inhibitors on natural killer cell activity in vitro [41]. It remains to be determined whether VA extracts interact with the PD-1 receptor on immunocompetent cells. Nevertheless, VA has been shown to stimulate γβ T cells [42] which have notable anti-tumor effects in several tumors, including NSCLC. These cells are also targeted by PD-1/PD-L1 inhibitors [43,44,45], potentially explaining why VA enhances the effect of PD-1/PD-L1 inhibitor therapy. Moreover, blocking PD-L1 on cancer cells or PD-1 on T cells with specific antibodies may create a microenvironment conducive to VA’s anti-cancer effect through other immunological mechanisms [46,47].

Primary resistance to PD-1/PD-L1 inhibitors in first- or second-line therapy for NSCLC ranges from 21% to 44%, but it could be lowered to 7% to 11% with the addition of chemotherapy [48]. This effect may be explained by the immunogenic cell death (ICD) induced by several chemotherapeutics [48]. During ICD, dying tumor cells release damage-associated molecular patterns, activating dendritic cells and initiating effective immune response against the tumor cells [49]. This renders the tumor cells more recognizable to the immune system, overcoming primary resistance to PD-1/PD-L1 inhibitors [49]. In relation to that, various antigens in VA extracts, such as MLs and viscotoxins, are potent modulators of cell types within the innate and adaptive immune systems. These antigens interact with toll-like receptors on antigen-presenting cells, with macrophages, natural killer cells, neutrophils, eosinophils, and both T and B cells [21,22]. VA extracts also enhance the maturation of dendritic cells, modulate immune cell and cytokine production [21,22,50,51,52], and promote anti-angiogenic [46,47,53] and pro-apoptotic processes [54]. Importantly, VA modulates rather than activates or inhibits immunocompetent cells, explaining its good tolerability and low incidence of severe side effects. Collectively, these mechanisms qualify VA for a role in inducing ICD and modification of the tumor microenvironment (TME). Immune checkpoint inhibitors reduce the immunosuppressive signals in the TME, allowing immune cells to infiltrate and attack the tumor more effectively [55]. VA extracts may further modify the TME to reduce tumor growth and immune cells accessibility [21,22,46,47,50,51,52,53,54]. Thus, in the present study, add-on VA could have been synergistically involved in ICD, modification of TME, and the overcoming of primary resistance to applied immune checkpoint inhibitors leading to longer overall survival in the NSCLC combination group.

Findings from recent RWD studies reveal that combining immune checkpoint blockade and VA therapy in advanced or metastatic NSCLC patients is associated with improved overall survival [17] and raises no safety concerns for VA [18,19,56]. The Phoenix-3 study also confirms that adverse event rates in NSCLC patients receiving either PD-1/PD-L1 inhibitors alone or with VA therapy did not differ significantly [56]. A recent review published by the British Society for Integrative Oncology and Imperial College London supports these findings, highlighting that data on immunotherapy-VA combinations show no safety signal and align with clinical experiences [57]. National guidelines for complementary therapies in oncological patients also indicate no evidence of an increased rate of serious adverse events from simultaneous use of immune checkpoint inhibitors and VA [20].

In summary, add-on VA therapy may enhance the effectiveness of immune checkpoint inhibitors, improving survival outcomes in advanced or metastatic NSCLC patients. This effect may be attributable to VA’s modulation of the TME, ICD induction, and immune system modulation, rendering tumor cells more accessible to immune attacks.

### Limitations and Strength

The non-randomized nature of this RWD study limits the results, including the potential for selection bias, as patient assignment to groups can be influenced by various characteristics, potentially skewing the assessment of the therapy’s efficacy. However, potential biases were addressed using multivariable logistic regression methods in the survival analyses to account for confounding factors. A key strength of this RWD study is its reflection of the actual use of PD-1/PD-L1 inhibitor therapy in NSCLC patients. It is the first to demonstrate a positive association between combined PD-1/PD-L1 inhibitor and abnobaViscum^®^ therapy with improved survival. As this study leverages RWD from an oncological registry, it allows for the inclusion of diverse patient populations, providing valuable insights into the effectiveness of integrative oncology approaches in real-world clinical settings.

## 4. Materials and Methods

### 4.1. Study Design

This study is an RWD analysis using information from a German Cancer Society-accredited source, the oncological registry Network Oncology (NO) [58], in line with the ESMO—Guidance for Reporting Oncological Real-World Evidence GROW criteria [31] (detailed in Appendix A). The design aims to evaluate the impact of integrative therapy with abnobaViscum^®^ on overall survival and associated factors in patients with advanced or metastatic non-small-cell lung cancer (NSCLC).

### 4.2. Study Population

Patients were included based on the following inclusion criteria: (i) documented first diagnosis of NSCLC (UICC stage III–IV) in the NO registry between July 2015 and May 2023, (ii) receipt of first-line immune checkpoint inhibitor therapy (anti-PD-1/PD-L1/CTL-4A) with or without abnobaViscum^®^, (iii) aged 18 years or older and of any gender, (iv) provided written informed consent.

### 4.3. Study Groups

Patients were allocated into two groups: (i) CTRL group: received standard oncological therapy, including PD-1/PD-L1 inhibitors without VA therapy; (ii) COMB group: received standard oncological therapy, including PD-1/PD-L1 inhibitors in combination with abnobaViscum^®^ therapy.

### 4.4. Study Setting and Administration of Therapy

The single-center study was conducted at a certified lung cancer center. During the observation period, (i) PD-1/PD-L1 inhibitors were administered according to standard clinical practice, (ii) abnobaViscum^®^ extracts were prescribed at the physician’s discretion, following the summary of product characteristics [59], and (ii) details of abnobaViscum^®^ therapy, including start and end dates, dosages, data of host tree, were documented. VA therapy included abnobaViscum^®^ (ABNOBA GmbH, Niefern-Öschelbronn, Germany) extracts and extracts from other producers (Helixor Heilmittel GmbH, Rosenfeld, Germany; Iscador AG, Arlesheim, Switzerland).

### 4.5. Data Collection

Data were extracted from the NO registry, including (i) demographic (age, gender, and other baseline characteristics), (ii) diagnosis and tumor details (tumor stage, histology, and molecular markers), (iii) treatment data (type, duration and combination of therapies), (iv) outcome data (survival status, follow-up information, tumor board and last contact data).

### 4.6. Follow-Up Data

Follow-up was routinely conducted six months after the initial diagnosis and annually in subsequent years. Loss to follow-up was defined as the absence of any follow-up visits.

### 4.7. Study Objectives

The primary objective was to evaluate overall survival (OS) in patients with advanced or metastatic NSCLC receiving anti-PD-1/PD-L1 treatment with and without abnobaViscum^®^ therapy. The secondary outcome aimed to descriptively assess whether specific variables were linked to a reduced risk of death.

### 4.8. Interdisciplinary Team

The multidisciplinary team of the presented study consisted of experts from various fields, including clinical practice, epidemiology, and biostatistics. This diversity of expertise was crucial in meeting the requirements of a successful RWD study according to the ESMO-GROW criteria [31]. Through close collaboration, we ensured that all aspects of the study were comprehensively addressed.

### 4.9. Ethics Issues

The study adheres to the principles outlined in the Declaration of Helsinki. Written informed consent was obtained from all patients prior to their enrollment. The study was approved by the ethics committee of the Medical Association Berlin (Eth-27/10).

### 4.10. Determination of Sample Size

To determine the necessary sample size for a two-sided test with an 80% power and a significance level of 5% using an allocation ratio of 0.8 (CTRL) to 0.2 (COMB) and an effect size of 0.6 [10,33], a total of 219 patients were required. This included 44 patients in the COMB and 175 patients in the CTRL group in order to confirm a statistically significant treatment effect, as outlined by Schoenfeld et al. [60].

### 4.11. Statistical Methods

Continuous variables were summarized using the median and interquartile range (IQR), while categorical variables were reported as absolute and relative frequencies. Distribution of continuous variables was visualized using histograms, Q-Q plot, and Shapiro–Wilk test. Patients with missing data were excluded from the analysis. Unpaired Mann–Whitney U Test or Student’s *t*-test for independent samples were used to compare continuous variables, and chi-square analysis was employed to compare categorical variables. All statistical tests were two-sided, and all analyses were exploratory in nature. Kaplan–Meier survival curves were generated for both the CTRL and the COMB group. Patient survival was calculated from the index date until the last recorded event including either the date of death, the last documented personal contact, the last interdisciplinary tumor board, or follow-up. For survival analyses, the index date was defined as the first date of start of anti-PD-1/PD-L1 therapy. Patients who had not died by the time of the analysis were censored. A year was defined as 365.25 days, and a month at 365.25/12 days.

To examine the influence of various factors on patient survival while minimizing potential confounding, we employed a multivariate stratified Cox proportional hazards model, adjusting for age, gender, tumor stage, ECOG performance status, PD-L1 status, and oncological treatment. Before conducting this analysis, we performed verification analyses to ensure that the proportional hazard assumptions were satisfied. All analyses were conducted using R-Studio version 2022.02.2 and R software version 4.1.2 (1 November 2021) “bird hippie”, which is a language and environment for statistical computing [35]. For Kaplan–Meier survival analysis and multivariate Cox proportional hazards analysis, we utilized the R package “survival” (version 3.5-5) [61]. The “prodlim” package was used for implementing nonparametric estimators for censored event history (survival) analysis (version 2019.11.13) [62]. To draw survival curves, the package “survminer” was used, version 0.4.9 [63]. The statistical analyses in this study not only encompass the outcomes but also take into account the internal and external validity of the data. We conducted sensitivity analyses such as subgroup analyses to verify the robustness of our results, to reduce potential biases, and to understand variations in the response.

## 5. Conclusions

Our findings indicate that the addition of abnobaViscum^®^ therapy is significantly associated with enhanced survival in patients being diagnosed with advanced or metastatic NSCLC receiving standard PD-1/PD-L1 inhibitor therapy, regardless of age, gender, metastatic status, or oncological treatment regimen (see summative Figure 4). Multivariate analysis revealed a 75% reduction in the adjusted hazard of death for PD-L1-positive patients receiving abnobaViscum^®^ therapy in combination with first-line PD-1 inhibitors (see summative Figure 4). VA’s bioactive compounds, including mistletoe lectins and viscotoxins, may complement immune checkpoint inhibitors by stimulating immune cells, inducing immunogenic cell death, and modifying the tumor microenvironment to improve immune accessibility. The findings underscore the role of abnobaViscum^®^ as a valuable complementary therapy that may enhance the efficacy of PD-1/PD-L1 inhibitors, possibly contributing to better survival outcomes in NSCLC. While these results highlight the clinical impact of add-on VA therapy, they should be further complemented with analyses of randomized controlled trials or registry-based randomized controlled trials.

## Figures and Tables

**Figure 1 pharmaceuticals-17-01713-f001:**
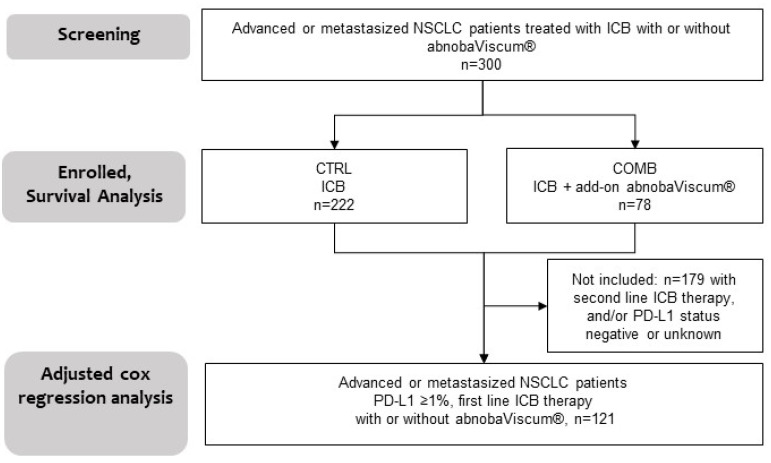
Study process flow. Patients with advanced or metastatic NSCLC who received PD-1/PD-L1 inhibitors, either with or without abnobaViscum^®^ therapy (*n* = 300), CRTL, received PD-1/PD-L1 inhibitors and no abnobaViscum^®^ therapy; COMB, received PD-1/PD-L1 inhibitors in conjunction with abnobaViscum^®^ therapy; ICB, immune checkpoint blockade; *n*, number; abnobaViscum^®^, abnobaViscum^®^ therapy; PD-L1 ≥1%, ≥1% tumor proportion score of programmed death-ligand 1.

**Figure 2 pharmaceuticals-17-01713-f002:**
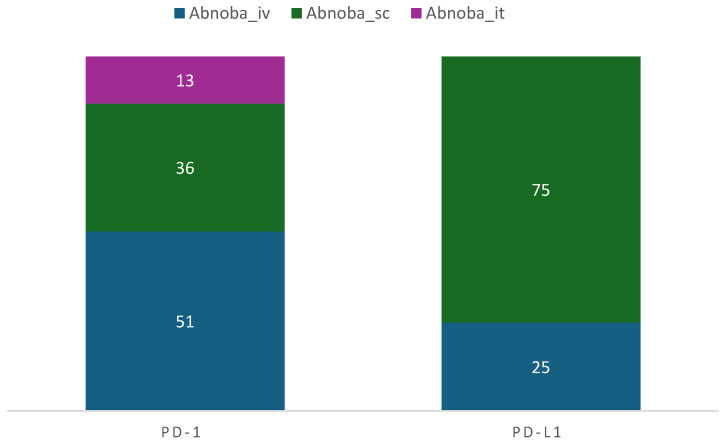
Characterization of combinational PD-1/PD-L1 inhibitor and abnobaViscum^®^ therapy. Abnoba, abnobaViscum^®^ therapy; iv, intravenous; sc, subcutaneous; it, intratumoral. Numbers are indicated as percent of patients.

**Figure 3 pharmaceuticals-17-01713-f003:**
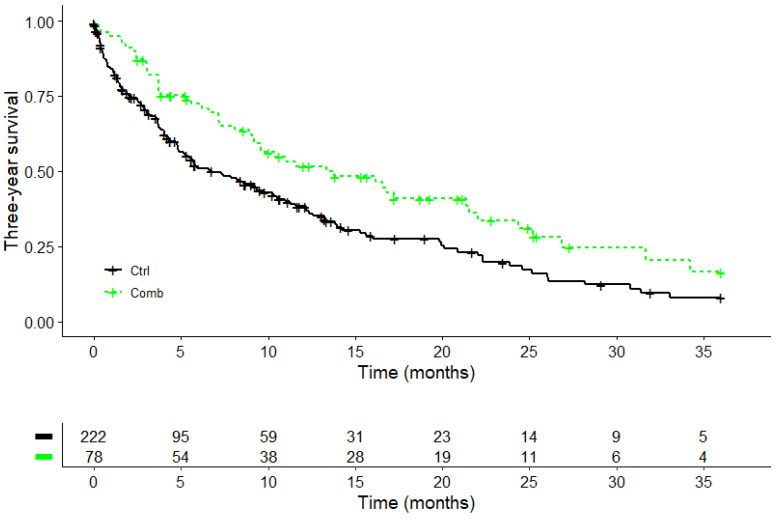
Kaplan–Meier survival curves displaying overall survival according to treatment in advanced or metastatic NSCLC (*n* = 300); Log-rank test: X^2^ = 7.9, *p* = 0.005; Ctrl, PD-1/PD-L1 inhibitors; Comb, PD-1/PD-L1 inhibitors + abnobaViscum^®^ therapy.

**Figure 4 pharmaceuticals-17-01713-f004:**
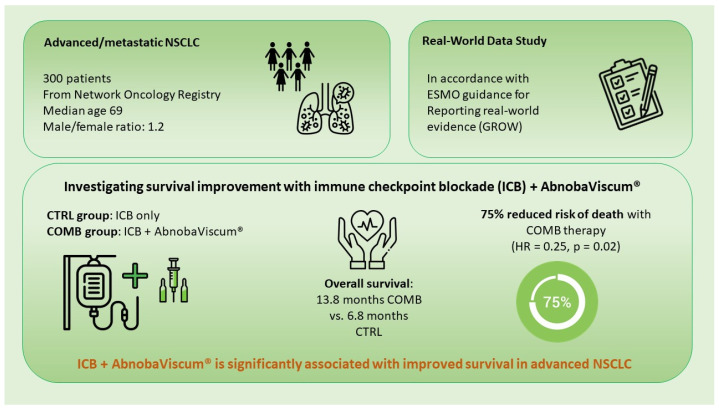
Summative figure of the study’s findings.

**Table 1 pharmaceuticals-17-01713-t001:** Characteristics of patients.

	Total Cohort (*n* = 300)		CRTL (*n* = 222)		COMB (*n* = 78)		*p*-Value
	N	%	N	%	N	%	
Age at first diagnosis, median (IQR) years	68	(62–76)	69.5	(63.5–76.8)	66.5	(60.3–74.0)	0.8971 ^(1)^
Age at first diagnosis, mean (SD) years	69.1	(10.0)	69.2	(9.4)	68.4	(12.2)	0.7822 ^(2)^
Gender							0.3053 ^(3)^
Female	137	45.7	97	43.7	40	51.3	
Male	163	54.3	125	56.3	38	48.7	
Histology							0.3023 ^(3)^
Non-squamous	189	63.0	141	63.5	48	61.5	
Squamous	90	30.0	63	28.4	27	34.6	
NSCLC, NA	21	7.0	18	8.1	3	3.8	
ECOG							0.5383 ^(3)^
0	68	22.7	50	22.5	18	23.1	
1	113	37.7	100	45.0	13	16.7	
2	39	13.0	33	14.9	6	7.7	
>3	28	6.7	20	9.0	8	10.3	
UICC stage							0.083 ^(3)^
UICC stage III	62	20.7	40	18.0	22	28.2	
UICC stage IV	238	79.3	182	82.0	56	71.8	

Patient characteristics. Please note that the percentages of sub-variables may not sum to 100% due to rounding. IQR referes to the interquartile range; CRTL indicates patients treated with PD-1/PD-L1 inhibitors alone, while COMB referes to patients receiving PD-1/PD-L1 inhibitors in conjunction with VA therapy. Distribution of age was visualized using a historgram, Q-Q plot and Shapiro–Wilk test. ^(1)^ Mann–Witney U Test. ^(2)^ Student’s *t*-test, ^(3)^ Chi-square test. UICC stands for Union International Contre le Cancer staging based on the UICC TNM classification system; ECOG represents the Eastern Cooperative Oncology Group.

**Table 2 pharmaceuticals-17-01713-t002:** Molecular characteristics of patient’s non-small-cell lung cancer.

	Total Cohort (*n* = 300)		CRTL (*n* = 222)		COMB (*n* = 78)		*p*-Value ^(1)^
	N	%	N	%	N	%	
PD-L1 status							0.007
PD-L1 status, positive/known	171/247	69.2	137/185	74.1	34/62	54.8	
PD-L1 status, ≥50 TPS/known	85/247	34.9	73	39.5	12	19.4	0.007
ALK translocation							1
negative/known	170/171	94.4	124/125	99.2	46/46	100	
EGFR (Exon 18–21) mutation							0.070
negative/known	136/148	91.9	106/112	94.6	30/36	83.3	
BRAF V600E-mutation							0.724
negative/known	154/157	98.1	114/117	97.4	40/40	100	

Molecular characteristics of patients with NSCLC. CRTL refers to patients receiving PD-1/PD-L1 inhibitors alone, while COMB denotes patients treated with PD-1/PD-L1 inhibitors plus VA. ^(1)^ Groups were compared using chi-square test; ALK, anaplastic lymphoma kinase; BRAF, B-rapidly accelerated fibrosarcoma; EGFR, epidermal growth factor receptor; PD-L1 programmed death ligand 1; TPS, tumor proportion score.

**Table 3 pharmaceuticals-17-01713-t003:** Characterization of antineoplastic therapy.

	Total Cohort (*n* = 300)		CRTL (*n* = 222)		COMB (*n* = 78)		*p*-Value ^(1)^
	N	%	N	%	N	%	
Radiation, bone	25	8.3	17	7.7	8	10.3	0.64
Radiation, brain	26	8.7	17	7.7	9	11.5	0.42
Radiation, primary tumor	33	11.0	21	9.5	12	15.4	0.22
Radiation, abdomen	1	0.3	1	0.5	0	0	1
Surgery	33	11.0	19	8.6	14	17.9	0.03
Chemotherapy	12	4.0	12	5.4	0	0	0.08
First-line immunotherapy	173	57.7	136	61.3	37	47.7	0.05
PD-L1/PD-1/CTL-A4 inhibitors							0.48
PD-L1 inhibitors	22	7.3	15	6.8	7	9.0	
PD-1 inhibitors	275	91.7	204	91.9	71	91.0	
CTL-A4 inhibitor	3	1.0	3	1.4	0	0	

Oncological therapy. N, number of patients; percentage, percent. CTL-A4, cytotoxic T-lymphocyte antigen 4; PD-L1, programmed death ligand; ^(1)^ Groups were compared using chi-square test; PD-1, programmed cell death protein 1; CRTL, patients receiving PD-1/PD-L1 inhibitors without VA therapy; COMB, patients receiving PD-1/PD-L1 inhibitors plus VA therapy.

**Table 4 pharmaceuticals-17-01713-t004:** Application and combination forms of add-on abnobaViscum^®^ therapy.

	Combination with Other VA (%)
abnobaViscum^®^ fraxini iv	37.2
abnobaViscum^®^ fraxini sc	19.2
abnobaViscum^®^ fraxini it	7.7
abnobaViscum^®^, other	4.2

Characterization of VA therapy. VA, Viscum album L.; %, percent; iv, intravenous; sc, subcutaneous; it, intratumoral; other, other host tree.

**Table 5 pharmaceuticals-17-01713-t005:** Median overall survival in patients with advanced or metastatic NSCLC in relation to treatment.

	N	Events	Median [Months]	95% CI [Months]
NSCLC, CTRL	222	138	6.8	4.9–10.4
NSCLC, COMB	78	49	13.8	9.2–22.0

Log rank test X2 = 7.9, *p* = 0.005

**Table 6 pharmaceuticals-17-01713-t006:** Factors associated with hazard of death.

	aHR	(95% CI)	*p*-Value	Adjusted *p*-Value ^(1)^
**abnobaViscum^®^ therapy vs. non-VA**	0.25	0.11–0.60	0.002 **	0.0153 *
**Age**	1.00	0.98–1.04	0.551	1.0000
**Female gender vs. male gender**	0.78	0.42–1.45	0.430	1.0000
**ECOG**	1.31	1.05–1.64	0.017 *	0.1372
**Surgery vs. no surgery**	0.84	1.19–0.33	0.726	1.0000
**UICC stage IV vs. stage III**	1.84	0.65–5.23	0.250	1.0000
**Brain radiation vs. no brain radiation**	0.99	0.39–2.47	0.975	1.0000
**PD-L1 TPS ≥50**	0.60	1.66–0.34	0.090	0.6362

Multivariate cox proportional analysis of factors linked to hazard of death in patients with advanced or metastatic PD-L1 positive NSCLC receiving first-line PD-1/PD-L1 inhibitors, *n* = 110, number of events 59, 11 observations deleted due to missing data; ^(1)^ adjusted *p*-value, *p*-value adjusted using the Bonferroni correction; aHR, adjusted hazard ratio of death; *, *p* < 0.05; **, *p* < 0.005; VA, abnobaViscum^®^.; UICC, UICC, union international contre le cancer; PD-L1, programmed death-ligand 1; TPS, tumor proportion score, Score (logrank) test = 21.45 on 8 df, *p* = 0.006.

## Data Availability

All relevant data are included in this manuscript.

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
