# Peer review of "Immune Checkpoint Blockade Combined with AbnobaViscum® Therapy Is Linked to Improved Survival in Advanced or Metastatic Non-Small-Cell Lung Cancer Patients: A Registry Study in Accordance with the ESMO Guidance for Reporting Real-World Evidence"

_pharmaceuticals, 2024, doi:10.3390/ph17121713_

Round 1
Reviewer 1 Report
Comments and Suggestions for Authors
The authors show in the presented work PD-1/PD-L1 Blockade Combined with AbnobaViscum® Therapy Is Linked to Improved Survival in Advanced or Metastatic NSCLC Patients, an ESMO-GROW Related RealWorld Data Registry Study, that in the RWD the efficacy of PD-1/PD-L1 inhibitor therapy in combination with abnobaViscum® therapy in patients with cancer was assessed. The presented results indicate that patients with advanced or metastatic non-small cell lung cancer who received PD-1/PD-L1 inhibitors in combination with abnobaViscum® therapy experienced better survival compared to patients who received PD-1/PD-L1 inhibitors alone.
Overall, the manuscript is well written. There are, however, a few minor comments to clarify:
1. In the statistical methods, the authors state that continuous variables were summarized using the median and interquartile range (IQR), but they did not state whether the variables were normally distributed and other statistical tests could have been used.
2. The table description does not specify the statistical test.
3. Table No. 3 does not specify which statistical test was used. The Bonferroni correction was not used in statistical calculations to prevent the problem of multiple comparisons.
Therefore, taking into account the above comments, I propose to make a correction to the manuscript.
Author Response
Comments of reviewer 1 and author’s answers
- The authors show in the presented work PD-1/PD-L1 Blockade Combined with AbnobaViscum® Therapy Is Linked to Improved Survival in Advanced or Metastatic NSCLC Patients, an ESMO-GROW Related RealWorld Data Registry Study, that in the RWD the efficacy of PD-1/PD-L1 inhibitor therapy in combination with abnobaViscum® therapy in patients with cancer was assessed. The presented results indicate that patients with advanced or metastatic non-small cell lung cancer who received PD-1/PD-L1 inhibitors in combination with abnobaViscum® therapy experienced better survival compared to patients who received PD-1/PD-L1 inhibitors alone. Overall, the manuscript is well written. There are, however, a few minor comments to clarify:
Author’s answer: Thank you for evaluating our manuscript as “well written”.
- In the statistical methods, the authors state that continuous variables were summarized using the median and interquartile range (IQR), but they did not state whether the variables were normally distributed and other statistical tests could have been used.
Author’s answer: Thank you for this point. We agree. We have clarified that age, as a continuous variable, follows a normal distribution. Additionally, we included visual and arithmetic tests in the methods section to evaluate this. Furthermore, Table 1 has been updated to include both tests for age: one assessing the median age and the other comparing mean age differences between the two groups.
- The table description does not specify the statistical test.
Author’s answer: Agree. We have, accordingly, included in the table 1, 2 and 3 the description the statistical tests.
- Table No. 3 does not specify which statistical test was used.
Author’s answer: Thank you for your comments. We have now indicated in table 3 (and also in table 1, 2 and 6) the description of the statistical tests.
- The Bonferroni correction was not used in statistical calculations to prevent the problem of multiple comparisons. Therefore, taking into account the above comments, I propose to make a correction to the manuscript.
Author’s answer: Thank you for your comments. We agree. We did not initially apply the Bonferroni correction because it is a conservative method that can increase the risk of Type II errors (false negatives), particularly in studies with a moderate to small sample size like ours. Our analysis was designed to identify meaningful associations rather than solely to rule out false positives, prioritizing sensitivity to detect potentially important predictors. Additionally, the primary aim of our Cox regression analysis was to evaluate specific, pre-defined variables selected based on prior literature or theoretical justification, rather than to conduct an exploratory analysis across numerous variables. This approach reduces the likelihood of spurious findings compared to purely exploratory methods.However, we have now applied the Bonferroni correction, as shown in Table 6. The adjusted p-value for the abnobaViscum® therapy is 0.0153, which remains statistically significant after the correction. Explanations have been added to both the Methods and Results sections.

Reviewer 2 Report
Comments and Suggestions for Authors
In this manuscript authors reported an ESMO-GROW study with 300 advanced or metastasized NSCLC patients receiving single PD-1/PD-L1 blockade treatment or combined with Viscum album L.(VA) treatment. They found the addition of VA therapy could significantly increase survival in patients comparing to single PD-1/PD-L1 treatment. The manuscript is well written and organized clearly. Authors presented the data and discussed the strength and limitations of this study. I would suggest acceptance for publication if authors address some minor issues.
1. In the Abstract section, Authors used ' three-year survival' term to compare patients survival time between groups. It would be more accurate to use 'overall survival' term.
2. In the Introduction section, the references order is not correct. 64 ,40, 46 after 9. Authors need to double check the references in the whole manuscript.
3. In Methods 2.5 section, authors stated CTRL and COMB ration was 0.2 and 0.8 respectively, however, in the study CTRL group patients was always more than COMB group.
4. In Table 1, the total cohhort n=415, authors did not mention this number in the manuscript. Where is this number from?
5. The format of 'Author contribution' need to be consistent with others.
Author Response
Comments of Reviewer 2 and author’s answers
In this manuscript authors reported an ESMO-GROW study with 300 advanced or metastasized NSCLC patients receiving single PD-1/PD-L1 blockade treatment or combined with Viscum album L. (VA) treatment. They found the addition of VA therapy could significantly increase survival in patients comparing to single PD-1/PD-L1 treatment. The manuscript is well written and organized clearly. Authors presented the data and discussed the strength and limitations of this study. I would suggest acceptance for publication if authors address some minor issues.
Author’s answer: Thank you for evaluating our manuscript as “well written”.
- In the Abstract section, Authors used ' three-year survival' term to compare patients survival time between groups. It would be more accurate to use 'overall survival' term.
Author’s answer: Thank you for this comment. We have changed, accordingly, from ,three-year survival, term to ,overall survival’.
- In the Introduction section, the references order is not correct. 64 ,40, 46 after 9. Authors need to double check the references in the whole manuscript.
Author’s answer: Thank you for this comment. We agree. We have checked and re-arranged the order of these and other references.
- In Methods 2.5 section, authors stated CTRL and COMB ratio was 0.2 and 0.8 respectively, however, in the study CTRL group patients was always more than COMB group.
Author’s answer: Thank you for detecting this typo. We have, accordingly, corrected it into 0.2 for COMB and 0.8 for CTRL.
- In Table 1, the total cohort n=415, authors did not mention this number in the manuscript. Where is this number from?
Author’s answer: Thank you for detecting this typo. We have corrected it. The total number is n=300.
- The format of 'Author contribution' need to be consistent with others.
Author’s answer: Thank you for this comment. We agree. The format has been checked and changed according to the format of the other supplementary information.

Reviewer 3 Report
Comments and Suggestions for Authors
The title should be concise and devoid of undefined acronyms. Moreover, specify PD1/PDL1 inhibitor.
There introduction section should be contains more information about lung cancer as global problem and its types and risk factors. Also, please add information about PD1/PDL1 inhibitors mechanism of action and examples. Moreover, add information about the active components of VA and thier mechanism in cytoxicity of lung cancer.
The experimental design should more clear and add more description of patients.
Please, check the manuscript for misuse of acronyms, each acronym should be completely written as first mentioned, no need at single or minimal uses of words.
The discussion section should be improved with additional similar published work.
Please check the manuscript for long sentences or paragraphs without references citation.
Please add the study limitations.
Comments on the Quality of English Language
Please check the manuscript for minor grammar errors and syntax.
Author Response
Comments of Reviewer 3 and author’s answers
- The title should be concise and devoid of undefined acronyms. Moreover, specify PD1/PDL1 inhibitor.
Author’s answer: Thank you for your comment. We have, accordingly, removed acronym such as NSCLC or GROW. In addition, we have changed “PD-1 and PD-L1” to ,immune checkpoint blockade, in order to avoid acronyms and a too lengthy title. However, we kept now only the acronym ,ESMO, (European Society for Medical Oncology) in the title, as this acronym is widely recognized within the oncology community, especially among clinicians and researchers. Including ESMO in the title adds credibility and highlights the study's association with a reputable international organization. We hope you agree.
- The introduction section should be contains more information about lung cancer as global problem and its types and risk factors. Also, please add information about PD1/PDL1 inhibitors mechanism of action and examples. Moreover, add information about the active components of VA and their mechanism in cytotoxicity of lung cancer.
Author’s answer: Thank you. We agree. We have now added more information about lung cancer as a global problem including types and risk factors. We have also tried to explain more the immune checkpoint inhibitor mechanisms of actions with relevant examples. In addition, we added more explanations on the active components of VA and their cytotoxic mechanisms, reflecting how they may act in lung cancer.
- The experimental design should more clear and add more description of patients.
Author’s answer: We agree, thank you for this comment. We have, accordingly, reorganized the Methods section into clearer subsections and provided additional details about the experimental design to enhance clarity." We have also added more descriptions to the patients’ inclusion and data.
- Please, check the manuscript for misuse of acronyms, each acronym should be completely written as first mentioned, no need at single or minimal uses of words.
Author’s answer: Thank you for this point. We have, in accordance to your comment, checked all acronyms and explained them when first mentioned. In addition we did not abbreviate when the word was used only once.
- The discussion section should be improved with additional similar published work.
Author’s answer: Thank you for your comment. We agree. We have, accordingly, tried to improve the discussion by rendering additional published work. However, as our effectiveness findings are one of the first for immune checkpoint blockade and VA, it was not possible for us to find other authors with similar published work. We hope you agree.
- Please check the manuscript for long sentences or paragraphs without references citation.
Author’s answer: We have checked and shortened long sentences or paragraphs and added reference citation, when needed.
- Please add the study limitations.
Author’s answer: Agree. We have, accordingly, added more study limitations.

Reviewer 4 Report
Comments and Suggestions for Authors
The manuscript titled "PD-1/PD-L1 Blockade Combined with AbnobaViscum® Therapy Is Linked to Improved Survival in Advanced or Metastatic NSCLC Patients, an ESMO-GROW Related Real- World Data Registry Study
1). The abstract is very long. It can be shortened.
2). There are several grammatical errors that need to be corrected.
"durvualumab" should be corrected to "durvalumab".
"useof" should be corrected to "use of".
"metastastatic" should be corrected to "metastatic".
"seÄ´ings" should be corrected to "settings".
"real-Word" should be corrected to "real-World".
"monocentric" should be clarified or replaced with a more common term like "single-center".
"Enrolment" should be corrected to "Enrollment".
3). Follow the same format for the tables throughout the manuscript.
4). Conclusions can be elaborated with more advantages.
Comments on the Quality of English LanguageThe language has to be improved.
Author Response
Comments of Reviewer 4 and author’s answers
The manuscript titled "PD-1/PD-L1 Blockade Combined with AbnobaViscum® Therapy Is Linked to Improved Survival in Advanced or Metastatic NSCLC Patients, an ESMO-GROW Related Real- World Data Registry Study
- The abstract is very long. It can be shortened.
Author’s answer: Thank you for your comment. We agree. We have, accordingly, shortened the abstract.
- There are several grammatical errors that need to be corrected.
"durvualumab" should be corrected to "durvalumab".
"useof" should be corrected to "use of".
"metastastatic" should be corrected to "metastatic".
"seÄ´ings" should be corrected to "settings".
"real-Word" should be corrected to "real-World".
"monocentric" should be clarified or replaced with a more common term like "single-center".
"Enrolment" should be corrected to "Enrollment".
Author’s answer: Agree. We have, in accordance to your suggestions, corrected the above mentioned words.
- Follow the same format for the tables throughout the manuscript.
Author’s answer: Thank you for this point. We agree and have, accordingly, followed the same format of the tables throughout the manuscript.
- Conclusions can be elaborated with more advantages.
Author’s answer: Thank you for your comment. We agree with you. We have, accordingly, elaborated the conclusions with more advantages.

Round 2
Reviewer 4 Report
Comments and Suggestions for Authors
Accept
Comments on the Quality of English LanguageBetter